# Resveratrol Targets Glycolytic Enzymes HK II and PKM2 to Promote Concurrent Apoptotic and Necrotic Cell Death in Malignant Melanoma

**DOI:** 10.3390/cimb47121006

**Published:** 2025-11-29

**Authors:** Yeji Lee, Sang-Han Lee, Dongsic Choi, Hae-Seon Nam, Ki Dam Kim, Min Hyuk Choi, Moon-Kyun Cho, Yoon-Jin Lee

**Affiliations:** 1Department of Biochemistry, College of Medicine, Soonchunhyang University, Cheonan 31511, Republic of Korea; yjyjlee37@naver.com (Y.L.); m1037624@sch.ac.kr (S.-H.L.); dongsicchoi@gmail.com (D.C.); 2Division of Molecular Cancer Research, Soonchunhyang Medical Research Institute, Soonchunhyang University, Cheonan 31511, Republic of Korea; namhs@sch.ac.kr; 3Department of Tropical Medicine, College of Medicine, Soonchunhyang University, Cheonan 31511, Republic of Korea; 4Department of Dermatology, Soonchunhyang University Hospital, Seoul 04401, Republic of Korea; 132351@schmc.ac.kr (K.D.K.); 146079@schmc.ac.kr (M.H.C.)

**Keywords:** resveratrol, apoptosis, necroptosis, malignant melanoma, hexokinase II, pyruvate kinase M2

## Abstract

Malignant melanoma exhibits high metastatic potential and resistance to chemotherapy, highlighting the need for novel therapeutic strategies. Resveratrol, a natural polyphenol, exerts anticancer effects by modulating cellular metabolism and apoptosis. In this study, we investigated its effects on hexokinase II (HK II) and pyruvate kinase M2 (PKM2) in G361 and SK-MEL-24 melanoma cells. Resveratrol reduced HK II and PKM2 expression and enzymatic activity, resulting in decreased ATP production and inhibition of glycolysis-dependent energy metabolism. Apoptosis was induced, as indicated by increased cleaved caspase-3, elevated Bax/Bcl-2 ratio, and enhanced caspase-3/7 activity. Necroptosis was also activated, evidenced by increased phosphorylation of RIP and MLKL. Cell cycle analysis revealed G0/G1 phase arrest, and Annexin V staining confirmed apoptosis. These effects were stronger in G361 cells than in SK-MEL-24 cells, suggesting that HK II- and PKM2-dependent metabolic traits influence resveratrol sensitivity. In summary, resveratrol activates both apoptotic and necroptotic cell-death pathways by inhibiting HK II and PKM2, highlighting its potential as a metabolism-targeted therapeutic agent for malignant melanoma.

## 1. Introduction

Malignant melanoma is an aggressive skin cancer originating from melanocytes in the epidermal basal layer. Its global incidence has risen steadily, especially in fair-skinned populations in Europe, North America, and Oceania, where both new cases and mortality continue to increase [1,2]. Although less common in Asian and African populations, patients in these regions are often diagnosed at more advanced stages, resulting in poorer outcomes [3]. Once distant metastasis occurs, prognosis is extremely poor, with median survival measured in months and 5-year survival rates below 5% [4,5]. Thus, early detection and more effective long-term treatment strategies are urgently needed.

Surgical excision is highly effective for localized melanoma; however, treatment of advanced disease requires complex multimodal approaches. Radiotherapy, cytotoxic chemotherapy, immune checkpoint inhibitors, and targeted molecular agents are widely employed but are associated with significant limitations, including systemic toxicity, immune-mediated adverse effects, and the frequent emergence of drug resistance [6,7]. Although immune checkpoint inhibitors targeting programmed death-1 (PD-1) or cytotoxic T-lymphocyte-associated protein-4 (CTLA-4) have improved outcomes for some patients, response rates vary widely, and treatment failure still occurs due to intrinsic non-responsiveness or acquired resistance [8,9,10]. Similarly, targeted inhibition of B-Raf proto-oncogene, serine/threonine kinase) (BRAF) signaling can induce initial tumor regression but often leads to metabolic rewiring and reactivation of survival pathways, ultimately promoting recurrence. These therapeutic shortcomings underscore the need for new strategies that target fundamental cellular processes essential for melanoma cell survival.

One promising approach involves exploiting metabolic vulnerabilities unique to tumor cells. Many melanoma cells undergo metabolic reprogramming and rely heavily on aerobic glycolysis—the Warburg effect—to sustain rapid proliferation, biomass accumulation, and survival within the tumor microenvironment. However, melanoma displays notable metabolic heterogeneity, with subsets of cells capable of switching between glycolysis and oxidative phosphorylation (OXPHOS) in response to nutrient availability or therapeutic pressure [11].

Hexokinase II (HK II), which phosphorylates glucose to glucose-6-phosphate in the first committed step of glycolysis, plays a pivotal role at the interface between energy metabolism and cell survival. HK II is frequently overexpressed in tumors and associates with the voltage-dependent anion channel (VDAC) on the mitochondrial outer membrane, thereby stabilizing mitochondrial integrity, preventing cytochrome c release, and suppressing intrinsic apoptosis [12,13]. Disruption of HK II expression or its mitochondrial binding reduces mitochondrial membrane potential, diminishes ATP production, and activates caspase-mediated apoptosis [14]. Another key glycolytic enzyme, pyruvate kinase M2 (PKM2), catalyzes the final step of glycolysis and regulates the balance between energy production and biosynthetic precursor generation. PKM2 also functions as a transcriptional co-activator influencing tumor growth, epithelial–mesenchymal transition, and drug resistance [15]. Elevated HK II and PKM2 expression thus represents a hallmark of metabolic adaptation in melanoma, linking altered glucose metabolism to tumor progression and therapy resistance.

Resveratrol (RSV) is a naturally occurring stilbene polyphenol found in grapes, red wine, peanuts, and various berries. Initially recognized for its cardioprotective effects, RSV is now known to exhibit diverse biological activities, including antioxidant, anti-inflammatory, and antitumor properties [16,17]. Previous studies have shown that RSV can inhibit cancer cell proliferation, induce cell-cycle arrest, and trigger programmed cell death across various tumor types [18,19]. Importantly, recent evidence suggests that RSV modulates mitochondrial function, alters HK II and PKM2 expression or localization, and regulates glycolytic flux, implying that metabolic control may represent a key mechanism underlying its anticancer effects [20]. Notably, RSV has been reported to concurrently suppress HK II and PKM2 in melanoma cells, thereby disrupting glycolytic energy production and promoting mitochondria-mediated cell death [21]. Despite these findings, the mechanistic relationship connecting RSV-mediated suppression of HK II and PKM2, mitochondrial dysfunction, and activation of apoptosis and necroptosis in melanoma remains incompletely understood. Furthermore, melanoma cell lines differ in metabolic reliance, and their responses to metabolic stressors such as RSV may vary substantially. Previous reports indicate that G361 and SK-MEL-24 melanoma cells differ in their metabolic dependencies, with G361 cells exhibiting a stronger reliance on glycolysis and reduced mitochondrial respiratory capacity, whereas SK-MEL-24 cells display greater OXPHOS activity and metabolic flexibility. These intrinsic metabolic differences may critically influence their susceptibility to glycolytic inhibition and mitochondrial stress [22]. Accordingly, utilizing two melanoma cell lines with distinct metabolic profiles provides a rationale for determining whether RSV-induced HK II and PKM2 suppression exerts differential effects depending on cellular metabolic reliance. This comparative approach strengthens the mechanistic interpretation of RSV’s metabolic targeting and improves the generalizability of the findings across heterogeneous melanoma subtypes.

The overarching aim of this study is to delineate how RSV-induced suppression of HK II and PKM2 orchestrates apoptotic and necroptotic cell death in melanoma, thereby establishing metabolic enzyme inhibition as a mechanistic foundation for RSV-mediated cytotoxicity. The experiments involving human tissue were undertaken to assess the preclinical relevance and potential translational applicability of the findings. Furthermore, comparisons with normal melanocytes were incorporated, and the metabolic distinctions between the two malignant melanoma cell lines are presented as evidence supporting the central research premise.

## 2. Materials and Methods

### 2.1. Tissue Preparation

Human skin specimens were obtained from patients who underwent surgery in the Department of Plastic and Reconstructive Surgery at Soonchunhyang University Bucheon Hospital, Korea, between December 2015 and November 2018. The collection and use of all human tissue samples were reviewed and approved by the Institutional Review Board of Seoul Soonchunhyang University Hospital (IRB No. 2018-04-007-002; approval date: 4 June 2018).

A total of six melanoma tissues (three male and three female patients) were collected and histologically confirmed by a board-certified pathologist. Normal skin tissues were harvested from the dorsal region of six female patients undergoing breast reconstruction using a latissimus dorsi flap. Immediately after surgical excision, all remaining tissue specimens were snap-frozen in liquid nitrogen and stored at –70 °C until subsequent Western blot analysis.

### 2.2. Reagents and Antibodies

RSV, dimethyl sulfoxide (DMSO), 3-(4,5-dimethylthiazol-2-yl)-2,5-diphenyltetrazolium bromide (MTT), 4′,6-diamidino-2-phenylindole (DAPI), paraformaldehyde, and β-actin antibody were purchased from Sigma-Aldrich (St. Louis, MO, USA). Primary antibodies against Cyclin D1 (92G2, #2978), Cyclin E1 (D7T3U, #20808), CDK4 (D9G3E, #12790), HK II (C64G5, #2867), PKM2 (D78A4, #4053), Bax (#5023), Bcl-2 (#2820), Cleaved Caspase-3 (Asp175, #9664), Caspase-3 (D3R6Y, #14220), Phospho-RIP (Ser166, D1L3S, #65746), RIP (D94C12, #3493), Phospho-MLKL (Ser358, D6H3V, #91,689), and MLKL (D2I6N, #14,993) were obtained from Cell Signaling Technology (Danvers, MA, USA). Horseradish peroxidase (HRP)-conjugated secondary antibodies, including anti-rabbit IgG (#7074) and anti-mouse IgG (#7076), were also obtained from Cell Signaling Technology. All antibodies were diluted at a ratio of 1:500 using 1× casein blocking solution (Catalog No. 37528; Thermo Fisher Scientific, Waltham, MA, USA) prior to use.

### 2.3. Cell Culture

Normal human epidermal melanocytes (HEMn-MP) were obtained from Cascade Biologics (Portland, OR, USA), and the human malignant melanoma cell lines G361 and SK-MEL-24 were purchased from the American Type Culture Collection (ATCC, Manassas, VA, USA). HEMn-MP cells were cultured in Medium 254 (M-254-500; Cascade Biologics) supplemented with 1% Human Melanocyte Growth Supplement to determine the concentration of RSV that does not exert cytotoxicity on normal cells. G361 and SK-MEL-24 cells were cultured in Dulbecco’s Modified Eagle’s Medium (DMEM; WelGene, Gyeongsan, Republic of Korea) containing 5% fetal bovine serum (FBS; Gibco, Gaithersburg, MD, USA), 100 U/mL penicillin, and 100 μg/mL streptomycin. All cell lines were incubated at 37 °C under a humidified atmosphere of 5% CO_2_ and 95% air.

### 2.4. Cell Viability Assay

The viability of HEMn-MP, G361, and SK-MEL-24 cells was evaluated using the MTT assay. Cells were seeded into 96-well plates (SPL Life Sciences, Pocheon, Republic of Korea) at a density of 1 × 10^4^ cells/mL and allowed to adhere for 24 h. The cells were then treated with resveratrol (RSV) at final concentrations of 0, 2.5, 5, 10, 20, 40, 60, 80, 100, and 120 μM for 48 h. Following treatment, 20 μL of MTT solution (0.1 mg/mL) was added to each well, and the plates were incubated for an additional 3 h at 37 °C in a 5% CO_2_ humidified incubator protected from light. After removing the medium, 200 μL of DMSO was added to each well to dissolve the formazan crystals, and the absorbance was measured at 550 nm using a microplate reader (GloMax-Multi Detection System; Promega, Madison, WI, USA). All viability experiments were performed in triplicate.

### 2.5. Cell Morphological Changes

To evaluate morphological alterations induced by RSV, G361 and SK-MEL-24 cells were seeded into 6-well culture plates at a density of 5 × 10^5^ cells/mL and allowed to adhere for 24 h in a 37 °C, 5% CO_2_ humidified incubator. The cells were then treated with RSV at concentrations of 0, 20, 40, and 60 μM and further incubated for 48 h. Cellular morphology was assessed, and representative images for each treatment concentration were acquired using a Leica EL6000 fluorescence microscope (Leica Microsystems GmbH, Wetzlar, Germany). All morphological analyses were conducted from three independent experiments.

### 2.6. DAPI Staining

DAPI staining was performed to assess nuclear condensation and fragmentation. G361 and SK-MEL-24 cells were seeded on 18-mm coverslips (Marienfeld GmbH, Lauda-Königshofen, Germany) in 6-well plates at 1 × 10^4^ cells/well and allowed to attach. Cells were treated with RSV (0, 20, 40, 60 μM) for 48 h, fixed with 4% paraformaldehyde for 20 min at room temperature, and dehydrated through a graded ethanol. Cells were then stained with DAPI (2 μg/mL) for 10 min in the dark. Nuclear morphology was imaged using a confocal fluorescence microscope (FluoView FV10i, Olympus, Tokyo, Japan). Experiments were performed in triplicate.

### 2.7. Annexin-V & Dead Cell Assay

To evaluate the effects of RSV on melanoma cell survival and apoptosis, G361 and SK-MEL-24 cells were seeded into 60-mm culture dishes at a density of 2.0 × 10^5^ cells/mL and allowed to adhere for 24 h. The cells were then treated with RSV at concentrations of 20, 40, and 60 μM for 48 h. Following treatment, the cells were harvested by trypsinization and stained using the Muse™ Annexin V & Dead Cell Kit (Luminex Corporation, Austin, TX, USA) according to the manufacturer’s protocol. Briefly, 100 μL of Muse™ Annexin V & Dead Cell Reagent was added to each sample, followed by incubation for 20 min at room temperature in the dark. Apoptotic cell populations, including live, early apoptotic, late apoptotic, and dead cells, were quantified using a Muse™ Cell Analyzer (Merck Millipore, Darmstadt, Germany). All apoptosis assays were performed in triplicate.

### 2.8. Cell Cycle Analysis

For cell cycle distribution analysis, G361 and SK-MEL-24 cells were treated with different concentrations of resveratrol (0, 20, 40, and 60 μM) for 48 h. Following treatment, the cells were detached with trypsin, collected, and washed twice with cold PBS. The cell pellets were then fixed in 70% ethanol at −20 °C until further analysis. Prior to staining, the ethanol-fixed cells were washed once with PBS and centrifuged at 300× *g* for 5 min. The pellets were gently resuspended in 200 μL of Muse™ Cell Cycle Reagent (Luminex Corp., Austin, TX, USA), protected from light, and incubated at room temperature for 30 min. After staining, cell cycle profiles were quantified using a Muse™ Cell Analyzer (Merck Millipore, Darmstadt, Germany) according to the manufacturer’s instructions. All cell cycle experiments were performed in triplicate.

### 2.9. Western Blot Analysis

G361 and SK-MEL-24 cells were exposed to RSV at concentrations of 20, 40, and 60 μM for 48 h. After treatment, the cells were washed with cold 1× PBS and lysed using RIPA buffer (1× PBS containing 1% NP-40, 0.5% sodium deoxycholate, 0.1% SDS, and 10 mg/mL phenylmethylsulfonyl fluoride). The total protein content of each sample was quantified with the Pierce^®^ BCA Protein Assay Kit (Thermo Fisher Scientific, Frederick, MD, USA). Equal amounts of protein (40 μg per lane) were separated on 4–12% NuPAGE polyacrylamide gels (Invitrogen, Carlsbad, CA, USA) and subsequently transferred to polyvinylidene fluoride (PVDF) membranes (Cytiva Life Sciences, Marlborough, MA, USA) at 30 V for 1 h. Membranes were blocked with 1× casein blocking buffer for 2 h at room temperature and incubated overnight at 4 °C with the respective primary antibodies. The next day, membranes were washed three times with 1× PBS containing 0.1% Tween-20 (PBST) and incubated with HRP-conjugated secondary antibodies for 1 h at room temperature. After washing under the same conditions, immunoreactive protein bands were visualized using an Enhanced Chemiluminescence (ECL) Detection Kit (Catalog No. W1001, Promega). Band intensity was quantified using ImageJ software (Version 1.0, NIH, Bethesda, MD, USA). Western blotting experiments were independently performed in triplicate. Densitometric analysis of the blots was carried out using ImageJ 1.0 software (National Institutes of Health, Bethesda, MD, USA).

### 2.10. ATP Content

After 48 h of RSV treatment, intracellular ATP levels in G361 and SK-MEL-24 cells were quantified using the CellTiter-Glo^®^ Luminescent Cell Viability Assay Kit (G9241, Promega Corporation, Madison, WI, USA) following the manufacturer’s guidelines. Prior to measurement, the cell culture plates were equilibrated to room temperature for approximately 30 min. Subsequently, 100 µL of CellTiter-Glo^®^ 2.0 reagent was added to an equal volume (100 µL) of culture medium in each well. The mixture was gently agitated on an orbital shaker for 2 min to ensure complete cell lysis and homogenization. The plates were then incubated at room temperature for an additional 10 min to stabilize the luminescent signal. Luminescence intensity, corresponding to relative intracellular ATP content, was recorded using a microplate luminometer. ATP measurements were obtained from three independent experiments.

### 2.11. Hexokinase Colorimetric Assay

Hexokinase (HK) activity was assessed using a commercial colorimetric assay kit (BioVision, Milpitas, CA, USA) according to the manufacturer’s protocol. RSV-treated G361 and SK-MEL-24 cells were lysed in HK assay buffer, centrifuged, and the supernatants were loaded onto a 96-well plate for measurement at 450 nm. HK activity was calculated following the kit’s instructions. HK activity assays were conducted in triplicate.

### 2.12. Caspase-3/7 Activity

Caspase-3/7 enzymatic activity was assessed in G361 and SK-MEL-24 cells following 48 h of RSV treatment using the ApoTox-Glo™ Triplex Assay Kit (G8090, Promega, Madison, WI, USA) in accordance with the manufacturer’s instructions. After treatment, an equal volume of Caspase-Glo^®^ 3/7 Reagent was added to each well, followed by gentle mixing and incubation at room temperature for 30 min to allow the luminescent signal to stabilize. Luminescence was then detected using a GloMax^®^ Multi Microplate Multimode Reader (Promega), and the relative caspase-3/7 activity was quantified based on the luminescence intensity. All caspase activity assays were performed in three independent replicates.

### 2.13. Wound Healing Assay

A wound healing assay was conducted to examine the effect of RSV on cell migration in G361 and SK-MEL-24 cells. The cells were seeded into 6-well plates at a density of 1 × 10^5^ cells/mL and cultured for 24 h to form a confluent monolayer. A linear scratch was then created across the center of each well using a sterile 10 μL pipette tip, after which the wells were gently washed with PBS to remove detached cells. Cells were subsequently treated with RSV at concentrations of 0, 20, 40, and 60 μM in medium supplemented with 5% FBS. To exclude the effects on cell proliferation and migration, treatments were also performed under serum-free conditions. After 48 h of incubation, wound closure was assessed using an inverted microscope (Leica Microsystems GmbH), and representative images were captured to quantify cell migration by measuring wound width. The wound healing assay was performed in triplicate.

### 2.14. Statistical Analysis

Statistical analysis was performed using GraphPad Prism software (version 9.5.1; GraphPad Software Inc., San Diego, CA, USA). All quantitative data are presented as the mean ± standard deviation (SD) from at least three independent experiments. Differences among groups were evaluated using one-way analysis of variance (ANOVA), followed by Tukey’s multiple comparison test.

## 3. Results

### 3.1. Comparison of HK II and PKM2 Expression in Normal Skin and Melanoma Tissues

The HK II and PKM2 was compared between normal skin tissues (*n* = 6) and malignant melanoma tissues (*n* = 6) using Western blot analysis. As shown in Figure 1A, melanoma specimens consistently exhibited higher HK II and PKM2 expression than normal skin. As shown in Figure 1B, the median (interquartile range) HK II expression in normal skin was 0.344 (0.250–0.456), whereas melanoma tissues exhibited a significantly higher median of 1.049 (0.715–1.219) (**** *p* < 0.0001). Similarly, the median (interquartile range) PKM2 expression in normal skin was 1.873 (1.390–2.567), while melanoma tissues showed a significantly increased median of 8.394 (7.147–9.540) (**** *p* < 0.0001).

### 3.2. Resveratrol Inhibited Cell Viability and Induced Apoptosis in Malignant Melanoma Cells

The cytotoxic effects of RSV on normal human epidermal melanocytes (HEMn-MP) and malignant melanoma cell lines (G361 and SK-MEL-24) were assessed using the MTT assay. The molecular structure of RSV used in this study is shown in Figure 2A. After 48 h of treatment with increasing concentrations of RSV (0–120 μM), cell viability was quantified (Figure 2B). In normal melanocytes, viability remained above 75% even at the highest concentration tested, indicating relatively low cytotoxicity. In contrast, melanoma cells exhibited a concentration-dependent reduction in viability.

G361 cells showed significant reductions in viability compared with the untreated control (*p* < 0.05, Figure 2B), with viability decreasing to 92.9%, 81.2%, 61.1%, and 44.6% at 5, 10, 20, and 40 μM RSV, respectively, and further to 27.8% at 120 μM. SK-MEL-24 cells also demonstrated a dose-dependent decrease, showing 91.1%, 79.3%, 70.8%, and 53.4% viability at the corresponding concentrations, with statistically significant reductions compared with control (* *p* < 0.05, Figure 2B). Based on the dose–response analysis, the IC_50_ values for G361 and SK-MEL-24 cells were calculated as 68.2 μM and 109.8 μM, respectively. Therefore, subsequent experiments were conducted using RSV concentrations ≤60 μM, which maintained over 90% viability in normal HEMn-MP melanocytes while effectively suppressing melanoma cell growth. Morphological observation confirmed a dose-dependent reduction in cell density and structural disruption, with G361 cells exhibiting more severe morphological alterations than SK-MEL-24 cells (Figure 2C).

To further determine whether RSV induces apoptotic cell death, nuclear morphologies were analyzed by DAPI staining. As shown in Figure 3A, RSV treatment (20–60 μM, 48 h) resulted in chromatin condensation and nuclear fragmentation in both melanoma cell lines, indicative of apoptosis. The proportion of cells displaying apoptotic nuclei increased in a concentration-dependent manner and was consistently higher in G361 cells compared to SK-MEL-24 cells (** *p* < 0.01, **** *p* < 0.0001). Apoptosis induction was quantitatively evaluated using Annexin V/7-AAD double staining (Figure 3B). In G361 cells, the total apoptotic population increased from 0.45% in the untreated group to 23.65%, 37.80%, and 47.72% following treatment with 20, 40, and 60 μM RSV, respectively (* *p* < 0.05, ** *p* < 0.01, **** *p* < 0.0001). In SK-MEL-24 cells, apoptosis increased from 0.15% in the control to 15.02%, 21.97%, and 29.63% at the corresponding concentrations (* *p* < 0.05, ** *p* < 0.01, **** *p* < 0.0001). These findings demonstrate that RSV induces apoptotic cell death in both melanoma cell lines, with G361 cells exhibiting greater sensitivity than SK-MEL-24 cells.

### 3.3. Effects of Resveratrol on the Cell Cycle of G361 and SK-MEL-24 Cell Lines

To evaluate the influence of RSV on cell cycle progression in malignant melanoma cells, cell cycle phase distribution (Figure 4A) and DNA content profiles (Figure 4B) were assessed using a Muse™ Cell Analyzer following 48 h of treatment. G361 and SK-MEL-24 cells were exposed to increasing concentrations of RSV (0, 20, 40, and 60 μM). In G361 cells, RSV treatment markedly increased the proportion of cells in the G0/G1 phase, with values of 55.1%, 60.5%, 63.0%, and 66.9%, respectively (* *p* < 0.05, *** *p* < 0.001, **** *p* < 0.0001), indicating a progressive G0/G1 arrest in response to RSV. Conversely, SK-MEL-24 cells exhibited only minor changes in the G0/G1 fraction (55.4%, 56.4%, 56.1%, and 58.1%), suggesting limited sensitivity to RSV-induced cell cycle modulation.

Consistent with these findings, the percentage of G361 cells in the S phase decreased in a concentration-dependent manner (18.57%, 17.63%, 16.87%, and 14.33%; * *p* < 0.05). However, SK-MEL-24 cells displayed comparatively modest alterations (26.10%, 21.07%, 18.17%, and 17.87%), again demonstrating a weaker response. A similar pattern was observed in the G2/M phase: G361 cells showed a stepwise reduction (17.50%, 17.50%, 16.93%, and 16.03%; ** *p* < 0.01, **** *p* < 0.0001), whereas SK-MEL-24 cells maintained relatively stable levels (26.37%, 26.10%, 26.33%, and 26.17%). To further determine whether RSV-mediated cell cycle arrest was associated with changes in regulatory proteins, Western blot analysis was performed (Figure 4C). In G361 cells, RSV reduced the expression of cyclin D1, cyclin E1, and CDK4 in a concentration-dependent manner after 48 h of exposure. In contrast, SK-MEL-24 cells showed no appreciable alterations in these protein levels under the same conditions. Collectively, these observations indicate that RSV suppresses proliferation primarily through G0/G1 phase arrest in G361 cells, whereas SK-MEL-24 cells exhibit comparatively lower sensitivity to RSV-induced cell cycle inhibition.

### 3.4. Resveratrol Regulates Hexokinase, Pyruvate Kinase M2, Apoptosis-Related Proteins, and Necroptosis in Malignant Melanoma Cells

To further elucidate the mechanism of RSV-induced cell death, we complemented cell cycle findings with protein-level analyses. Western blotting was performed to evaluate the effects of RSV on key glycolytic enzymes, HK II and PKM2, both of which are crucial for maintaining glycolytic flux and energy homeostasis. In G361 melanoma cells, RSV treatment led to a concentration-dependent decrease in the expression of both HK II and PKM2, indicating suppression of glycolysis-associated energy metabolism. In contrast, SK-MEL-24 cells showed minimal change in HK II expression and only a slight reduction in PKM2, suggesting cell line-specific differences in metabolic sensitivity to RSV.

Next, apoptosis-associated proteins were analyzed, including caspase-3, cleaved caspase-3, Bax, and Bcl-2. In G361 cells, RSV treatment resulted in a reduction in caspase-3 and Bcl-2 levels, along with an elevation in Bax and cleaved caspase-3 expression, reflecting activation of apoptotic signaling. By contrast, SK-MEL-24 cells displayed minimal alterations in the same markers, indicating a more limited apoptotic response (Figure 5A). Additionally, necroptosis signaling was assessed by determining phosphorylated forms of MLKL and RIP. A dose-dependent increase in p-MLKL and p-RIP was observed following RSV treatment in both melanoma cell lines; however, the magnitude of induction was notably higher in G361 cells (Figure 5B).

Consistent with the protein expression results, intracellular ATP levels decreased as RSV concentration increased. Compared with the control, ATP levels were reduced to 71.23%, 62.86%, and 53.90% in G361 cells, and to 84.13%, 78.93%, and 70.23% in SK-MEL-24 cells (* *p* < 0.05, ** *p* < 0.01, *** *p* < 0.001, **** *p* < 0.0001; Figure 5C). HK enzymatic activity also showed a marked dose-dependent reduction in G361 cells (74.99%, 59.43%, and 28.38% of control; **** *p* < 0.0001), while SK-MEL-24 cells exhibited no significant changes (Figure 5D). In contrast, caspase-3/7 enzymatic activity increased in response to RSV, rising to 120.8%, 136.5%, and 173.7% in G361 cells (** *p* < 0.01, *** *p* < 0.001, **** *p* < 0.0001), and to 104.3%, 107.7%, and 113.8% in SK-MEL-24 cells, with the increase being substantially greater in G361 cells (Figure 5E).

### 3.5. Effects of Resveratrol on the Migration of Malignant Melanoma Cells

To evaluate the effect of RSV on the migratory capacity of malignant melanoma cells, G361 and SK-MEL-24 cells were treated with RSV at concentrations of 0, 20, 40, and 60 µM for 48 h. Wound healing assays were performed using both 5% FBS-supplemented medium and serum-free medium to exclude the effects of cell proliferation and migration (Figure 6). The wound gap was monitored immediately after scratching (0 h) and at 48 h to quantify cell migration. Both G361 and SK-MEL-24 cells exhibited a marked, dose-dependent reduction in wound closure in response to RSV. The inhibitory effect of RSV on wound healing was more pronounced in G361 cells than in SK-MEL-24 cells. Moreover, the decrease in wound closure induced by RSV was greater under serum-free conditions than in 5% FBS-containing medium, indicating that the anti-migratory response to RSV varies depending on the characteristics of the melanoma cell line (* *p* < 0.05, ** *p* < 0.01, **** *p* < 0.0001).

## 4. Discussion

Malignant melanoma is characterized by profound metabolic reprogramming that enables rapid proliferation, survival under hypoxic and nutrient-deficient microenvironments, and resistance to conventional apoptosis-based therapies. In particular, melanoma cells frequently exhibit enhanced aerobic glycolysis coupled with mitochondrial functional adaptation, allowing them to maintain ATP synthesis and redox balance while simultaneously evading programmed cell death [23]. In this context, identifying molecular regulators that serve as nodal points linking energy metabolism and cell survival has become a central objective in melanoma therapeutics. The present study demonstrates that RSV disrupts melanoma cell survival by directly suppressing HK II, a pivotal enzyme that integrates glycolytic ATP production with mitochondrial anti-apoptotic signaling, as well as PKM2, which catalyzes the terminal step of glycolysis and regulates anabolic metabolism and tumor growth. Through HK II and PKM2 downregulation, RSV induced a marked reduction in intracellular ATP levels and triggered both intrinsic apoptosis and necroptosis in G361 and SK-MEL-24 melanoma cells, accompanied by reduced cell migration capacity (Figure 7).

HK II catalyzes the conversion of glucose to glucose-6-phosphate, representing the first and rate-limiting step of glycolysis. On the other hand, PKM2 converts phosphoenolpyruvate to pyruvate in the final glycolytic step and is known to act as a metabolic switch that promotes cancer cell proliferation, redox balance, and gene transcription under oncogenic signaling [24]. Elevated expression of both HK II and PKM2 represents a hallmark of metabolic reprogramming in melanoma and contributes to enhanced glycolytic flux and survival under stress conditions. Elevated HK II expression is a well-recognized hallmark of metabolic reprogramming in melanoma and other aggressive tumors [25]. Importantly, HK II localizes to the outer mitochondrial membrane through binding to the VDAC, a structural interaction that restricts mitochondrial outer membrane permeabilization and prevents cytochrome c release [13,26]. Thus, HK II supports melanoma survival through both metabolic fueling and direct inhibition of apoptosis. Similarly, PKM2 not only sustains glycolytic ATP production but also regulates transcriptional programs linked to proliferation and metabolic adaptation, further supporting tumor progression [21,27].

Our findings reveal that RSV treatment significantly decreases HK II protein levels and enzymatic activity, along with a concomitant reduction in PKM2 expression and activity, resulting in a pronounced ATP reduction in G361 cells. By contrast, SK-MEL-24 cells exhibited a weaker metabolic response, indicating intrinsic heterogeneity in glycolytic dependency. This differential response between G361 and SK-MEL-24 cells likely reflects their metabolic phenotypes: G361 cells appear to be highly glycolysis-dependent, rendering them more vulnerable to HK II/PKM2 inhibition, whereas SK-MEL-24 cells likely possess greater metabolic flexibility, enabling partial compensation through oxidative phosphorylation (OXPHOS) [28]. This supports the emerging view that melanoma subtypes range from highly glycolysis-dependent phenotypes to more metabolically flexible phenotypes capable of shifting toward oxidative phosphorylation under stress [29,30]. The stronger RSV sensitivity of G361 cells therefore reflects a metabolic state in which both HK II and PKM2 act as essential determinants of glycolytic flux and survival.

The decline in ATP levels following HK II suppression is closely linked to the induction of intrinsic apoptosis. In both melanoma cell lines, RSV treatment elevated the Bax/Bcl-2 ratio, promoting mitochondrial outer membrane permeabilization and facilitating cytochrome c release [31,32]. Increased levels of cleaved caspase-3 and heightened caspase-3/7 activity further substantiate the activation of mitochondrial apoptosis. These data align with earlier reports that RSV can induce mitochondrial dysfunction, promote ROS generation, and destabilize mitochondrial membrane potential in melanoma cells [33,34]. However, a major novel finding of the present study is the concurrent activation of necroptosis, characterized by phosphorylation of RIP and MLKL. Necroptosis is a regulated, kinase-driven form of cell death that becomes prominent when caspase signaling is limited or inefficient [35,36]. Since melanoma frequently acquires resistance to apoptosis through overexpression of anti-apoptotic Bcl-2 family members and other survival regulators [37], the ability of RSV to induce necroptosis is mechanistically and therapeutically significant. The results indicate that metabolic disruption may create cell-death-permissive conditions that allow a switch from apoptosis to necroptosis, thereby overcoming intrinsic apoptosis resistance that is a hallmark of aggressive melanoma.

An important translational implication of this study is that both HK II and PKM2 may serve as predictive biomarkers and therapeutic targets in melanoma. Since G361 cells exhibited stronger responses to RSV-induced suppression, assessing glycolytic dependency may allow clinicians to stratify melanoma tumors into treatment-responsive and treatment-resistant phenotypes. Moreover, the findings support mechanistic synergy with current targeted therapies. Melanomas harboring BRAF V600E mutations, when treated with BRAF inhibitors, frequently undergo metabolic rewiring that increases glycolysis and HK II/PKM2 dependency [38]. Therefore, inhibition of these enzymes could enhance therapeutic response and delay resistance to BRAF/MEK inhibitors. Furthermore, necroptosis is increasingly recognized as an immunogenic form of cell death that promotes dendritic cell recruitment and enhances anti-tumor T-cell priming [39,40]. Thus, RSV-induced necroptosis could potentially augment the clinical effectiveness of immune checkpoint blockade therapies including anti-PD-1 and anti-CTLA-4 antibodies.

In addition to inducing cell death, RSV significantly inhibited melanoma cell migration. Metastatic spread is the primary determinant of melanoma mortality [41], and suppression of motility represents a critical therapeutic benefit. RSV has previously been reported to inhibit epithelial-to-mesenchymal transition (EMT), reduce matrix metalloproteinase (MMP) expression, and disrupt cytoskeletal dynamics [42,43]. The present results confirm that RSV reduces migration in both G361 and SK-MEL-24 cells, with a stronger reduction observed in the more glycolysis-dependent G361 line. These data suggest that the anti-migratory effects of RSV may be metabolically mediated, reflecting the central role of HK II and PKM2 in ATP generation, cytoskeletal remodeling, and focal adhesion turnover.

Another clinically meaningful consideration is the delivery efficiency of RSV. Although RSV is intrinsically characterized by rapid metabolic turnover and limited systemic persistence [44], recent progress in pharmaceutical engineering has substantially enhanced its translational promise. Nanotechnology-based formulation platforms—such as polymeric nanoparticles, liposomal encapsulation, and stimuli-responsive delivery systems—have demonstrated improved tumor-targeted accumulation, prolonged circulation half-life, and reduced nonspecific clearance [34,45]. These advancements provide a practical foundation for applying RSV in therapeutic contexts that require sustained metabolic perturbation within tumor tissues.

In parallel, the molecular interface between RSV and HK II as well as PKM2 warrants further mechanistic refinement. HK II function is regulated not only at the expression level but also through post-translational modifications and mitochondrial membrane association, which together determine its catalytic efficiency and pro-survival signaling capacity. PKM2, likewise, is regulated by phosphorylation, acetylation, and dimer–tetramer interconversion, processes that control its metabolic and non-metabolic functions in tumor cells. Therefore, determining whether RSV influences these regulatory mechanisms or disrupts interactions between HK II, PKM2, and mitochondrial proteins such as VDAC will be essential to fully elucidate how RSV orchestrates metabolic remodeling and cell death execution in melanoma. Such mechanistic clarity would further support the strategic development of metabolic-targeted therapies leveraging RSV or HK II/PKM2-directed interventions.

## 5. Conclusions

In summary, these results demonstrate that RSV exerts its antitumor effects by inducing mitochondrial-specific apoptosis throughout the entire process in melanoma cells. Specifically, RSV inhibited ATP production and activated intracellular apoptotic and necrotic signaling, particularly in combination with HK II and PKM2. Furthermore, G361 and SK-MEL-24 cell antagonists induced massive migration capacity in response to RSV. Therefore, strategies targeting the same processes as HK II and PKM2 may be promising therapeutic approaches for overcoming the versatility and apoptosis resistance of melanoma.

## Figures and Tables

**Figure 1 cimb-47-01006-f001:**
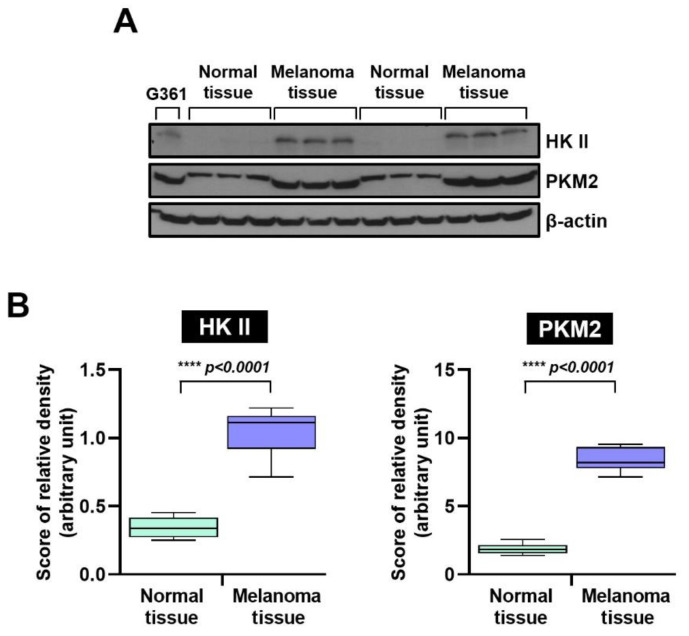
Hexokinase II and PKM2 protein expression in human normal skin and malignant melanoma tissues, and relative protein expression. (**A**) Western blot analysis. Levels of HK II and PKM2 were markedly higher in melanoma tissues than in normal tissues. β-actin served as a loading control. (**B**) Quantitative comparison of relative HK II and PKM2 protein levels in normal (*n* = 6) and melanoma tissues (*n* = 6). Statistical significance was determined using Tukey’s post hoc multiple comparison tests (**** *p* < 0.0001). HK II: Hexokinase II, and pyruvate kinase M2: PKM2.

**Figure 2 cimb-47-01006-f002:**
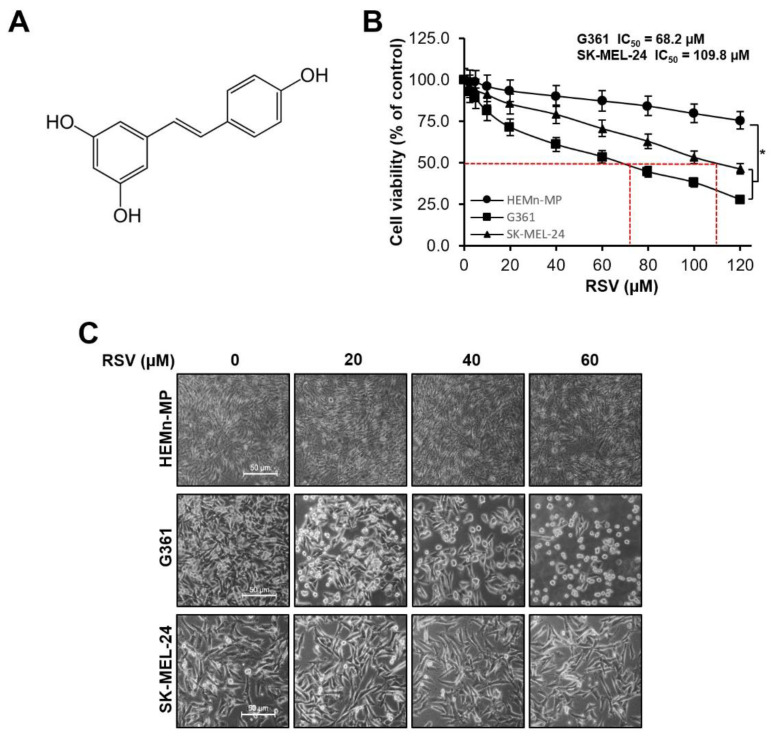
Effects of resveratrol on cell viability and morphology in malignant melanoma cells. (**A**) Chemical structure of resveratrol (RSV). (**B**) Cell viabilities of non-tumorigenic human epidermal melanocytes (HEMn-MP) and malignant melanoma cell lines (G361 and SK-MEL-24) after 48 h of RSV treatment, assessed by MTT assay. Results are presented as means ± Standard error (SE) of three independent experiments (* *p* < 0.05). (**C**) Representative phase-contrast images showing morphological alterations in HEMn-MP, G361, and SK-MEL-24 cells following RSV exposure for 48 h (scale bar = 50 μm; original magnification ×400). RSV: resveratrol.

**Figure 3 cimb-47-01006-f003:**
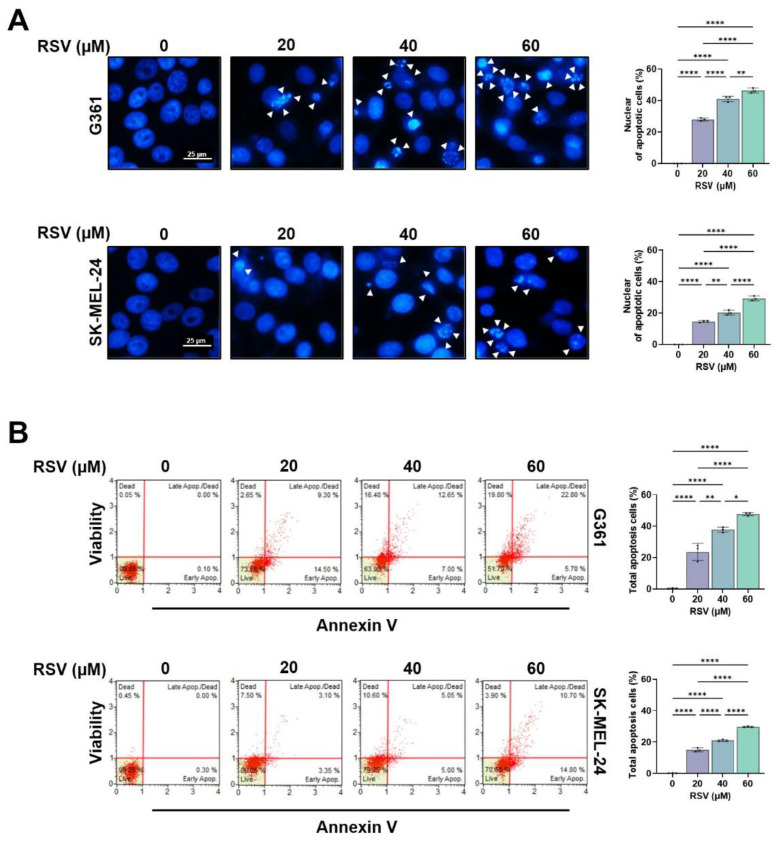
Resveratrol induces apoptotic and necroptotic cell death in malignant melanoma cells. G361 and SK-MEL-24 cells were treated with RSV (20, 40, 60 μM) for 48 h. (**A**) Representative fluorescence images of DAPI-stained nuclei (scale bar = 25 μm; original magnification ×200). Condensed or fragmented nuclei, indicative of apoptosis, are indicated by white arrows. (**B**) Quantification of apoptotic cell populations by Annexin-V/7-AAD staining using a Muse™ Cell Analyzer. Annexin-V-positive/7-AAD-negative cells were classified as early apoptotic, whereas Annexin-V-positive/7-AAD-positive cells were classified as late apoptotic. Data are presented as mean ± SE (*n* = 3). Statistical significance: * *p* < 0.05, ** *p* < 0.01, **** *p* < 0.0001. RSV: resveratrol.

**Figure 4 cimb-47-01006-f004:**
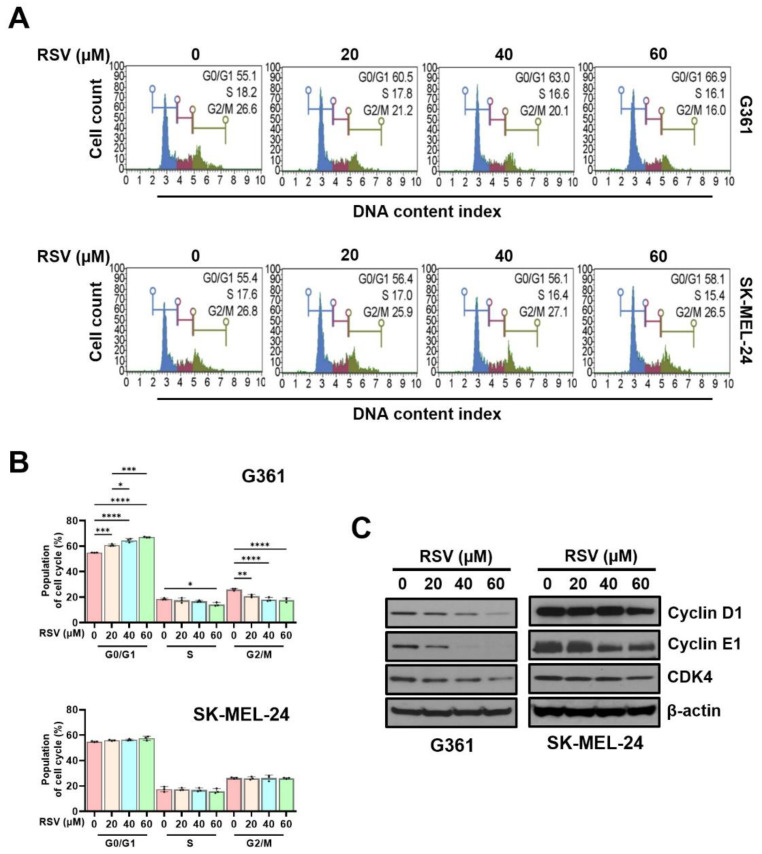
Resveratrol induced G0/G1 cell cycle arrest in malignant melanoma cells. G361 and SK-MEL-24 cells were treated with RSV (0, 20, 40, 60 µM) for 48 h. (**A**) Representative histograms of cell cycle distribution following staining with Muse™ Cell Cycle Reagent and analysis on a Muse™ Cell Analyzer. (**B**) Quantitative analysis of the percentage of cells in G0/G1, S, and G2/M phases. Data represent mean ± SE (*n* = 3). (**C**) Western blot analysis of cell cycle regulatory proteins (cyclin D1, cyclin E1, and CDK4) after 48 h of RSV treatment. β-Actin served as a loading control. Relative band intensities were quantified against the control. Statistical significance: * *p* < 0.05, ** *p* < 0.01, *** *p* < 0.001, **** *p* < 0.0001 (post hoc test). RSV: resveratrol.

**Figure 5 cimb-47-01006-f005:**
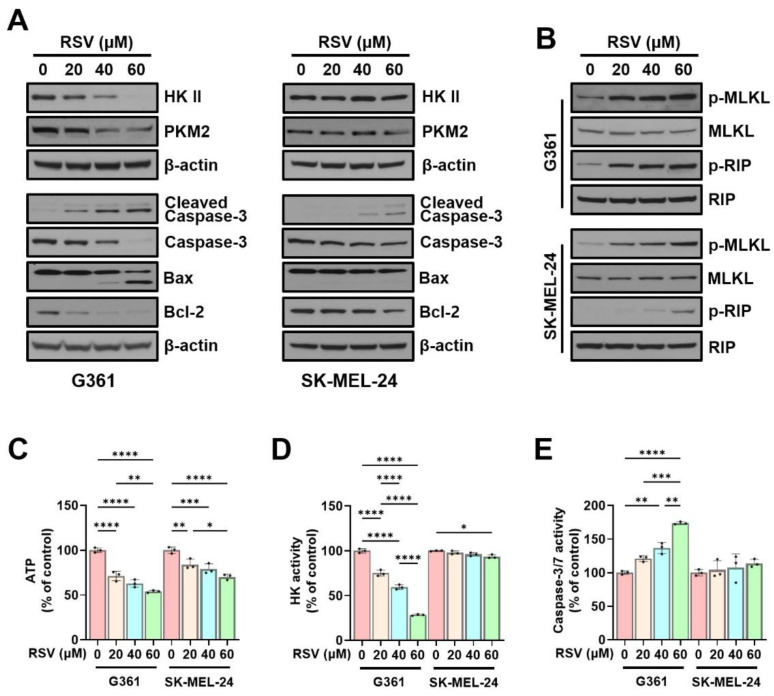
Resveratrol modulates glycolytic enzymes and induces apoptosis and necroptosis in malignant melanoma cells. G361 and SK-MEL-24 cells were treated with RSV (0, 20, 40, 60 µM) for 48 h. (**A**) Western blot analysis of HK II, PKM2, and apoptosis-related proteins (cleaved caspase-3, caspase-3, Bax, and Bcl-2). β-Actin served as loading control. (**B**) Western blot analysis of necroptosis-associated proteins, including phosphorylated MLKL (p-MLKL) and phosphorylated RIP (p-RIP). Total MLKL and total RIP were used as loading controls. (**C**) Intracellular ATP levels. (**D**) Hexokinase enzymatic activity. (**E**) Caspase-3/7 activity. Data are expressed as mean ± SE (*n* = 3). Statistical significance was determined by one-way ANOVA followed by Tukey’s post hoc test (* *p* < 0.05, ** *p* < 0.01, *** *p* < 0.001, **** *p* < 0.0001). HK II: Hexokinase II; PKM2: pyruvate kinase M2; RSV: resveratrol.

**Figure 6 cimb-47-01006-f006:**
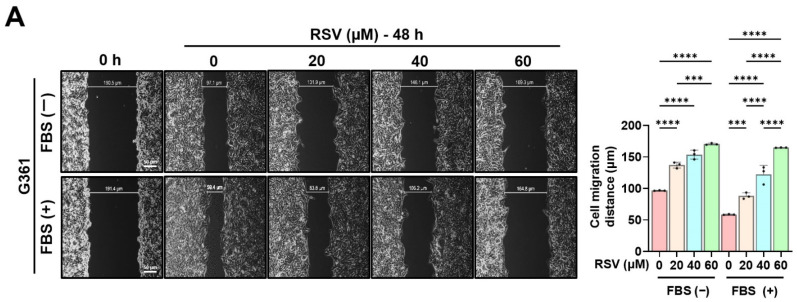
Resveratrol suppresses the migratory ability of malignant melanoma cells. A wound healing assay was performed in G361 (**A**) and SK-MEL-24 (**B**) cells treated with RSV (0, 20, 40, 60 μM) for 48 h. Assays were conducted under both 5% FBS-supplemented and serum-free conditions to minimize confounding effects of proliferation. Wound areas were imaged at 0 h and 48 h to assess cell migration (scale bar = 50 μm; original magnification ×400). Data are presented as mean ± SE (*n* = 3). Statistical significance: * *p* < 0.05, ** *p* < 0.01, *** *p* < 0.001, **** *p* < 0.0001 (post hoc test). RSV: resveratrol.

**Figure 7 cimb-47-01006-f007:**
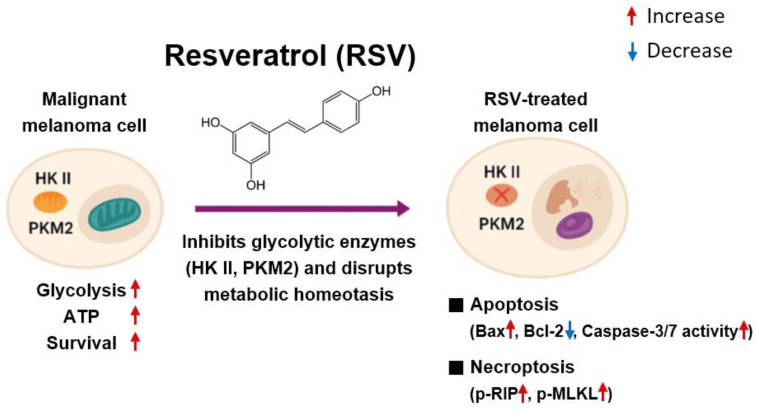
Proposed mechanism of resveratrol-induced cell death in malignant melanoma cells. Resveratrol decreases HK II and PKM2 expression, resulting in reduced glycolysis-dependent ATP production. The subsequent decline in cellular energy availability induces mitochondrial dysfunction and promotes intrinsic apoptosis through an increased Bax/Bcl-2 ratio and activation of caspase-3/7. Simultaneously, metabolic stress triggers necroptosis through phosphorylation of RIP, and MLKL. These coordinated mechanisms collectively contribute to cell death in melanoma cells.

## Data Availability

All data produced or assessed during this study are included in this article.

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
