# Peer review of "Resveratrol Targets Glycolytic Enzymes HK II and PKM2 to Promote Concurrent Apoptotic and Necrotic Cell Death in Malignant Melanoma"

_cimb, 2025, doi:10.3390/cimb47121006_

Round 1
Reviewer 1 Report
Comments and Suggestions for Authors
1. Abstract:
It would be advisable to shorten some of the longer sentences or simplify the list of markers to improve readability. The research aim should also be stated more clearly. Overall, the writing style is appropriate, although the sentences are often complex. I suggest breaking some of them into shorter units to avoid monotony. For example, the last sentence of the abstract could be divided and made more concise.
2. Introduction:
While the introduction provides valuable context, it is at times overly detailed. It could be shortened by removing some of the extensive epidemiological information and focusing more directly on the main objective of the study, which should be clearly articulated at the end of the section.
3. Methods:
-
A clear statement regarding the number of replicates for each assay should be added directly in the experimental descriptions. Currently, this information appears only in the statistical section or in figure legends, which may make the methodology harder to follow.
-
Although the level of detail demonstrates precision, the methodological section is very lengthy. For consideration: certain parts of manufacturers’ instructions (e.g., kit procedures, the formula for calculating hexokinase activity) could be summarized or replaced with a reference such as “according to the manufacturer’s protocol,” as is common practice in publications.
4. Results:
It would be helpful to include comments on the statistical significance of the main differences observed (e.g., p-values or significance thresholds).
The only minor terminological issue concerns the use of the term “necrosis” in the title of section 3.4, whereas the study actually investigated necroptosis. Using the term “necroptosis” would avoid confusion with accidental necrosis. This does not affect the conclusions but is worth correcting in the final version.
For Figure 2C, please provide higher-quality microscopic images.
Please clarify why, after determining the IC50 values in the cytotoxicity assessment, these values were not used for subsequent experiments and why different concentrations of RSV were selected instead.
I also recommend moving the wound healing assay results to the Results section and removing them from the Discussion.
Some abbreviations were not explained when first used (e.g., RSV, VDAC—it is worth ensuring that they are defined earlier).
Comments on the Quality of English LanguageTypos and editorial errors:
“Tropocal Medicine” (p. 1, affiliations) → should be “Tropical Medicine.”
“reserch” in Acknowledgments (p. 16) → should be “research.”
“left-right motility” in Conclusions (p. 15) → this expression is unclear. It probably meant “reduced cell motility,” but the form “left-right motility” is incomprehensible and should be clarified or replaced, e.g., with “cell motility” or “migration capacity.”
Author Response
Dear Reviewers:
We sincerely thank you for the opportunity to revise our manuscript and appreciate the time and effort the reviewers have dedicated to evaluating our work.
The reviewers’ comments were very helpful in improving the clarity and quality of our manuscript. We have carefully addressed all the suggestions and made revisions accordingly. We believe that these changes have substantially improved the manuscript, and we hope that the revised version is now suitable for publication in Current Issues in Molecular Biology.
All modifications have been made in accordance with the reviewers’ recommendations, and the revised sections have been highlighted in red for clarity.
Thank you for your consideration.
Sincerely yours,
Yoon-Jin Lee, PhD
Department of Biochemistry, College of Medicine,
Soonchunhyang University, Cheonan 31511, Republic of Korea
Phone: +82 41 570 2443
E-mail: leeyj@sch.ac.kr
Point-by-Point Responses to Editor
Manuscript ID: cimb-4015648
Title: Resveratrol Targets Glycolytic Enzymes HK II and PKM2 to Promote Concurrent Apoptotic and Necrotic Cell Death in Ma-lignant Melanoma
Authors: Yeji Lee 1, Sang-Han Lee 1,2, Dongsic Choi 1,2, Hae-Seon Nam 2,3, Ki Dam Kim 4, Min Hyuk Choi 4, Moon-Kyun Cho 2,4*,† and Yoon-Jin Lee 1,2*,†
1 Department of Biochemistry, College of Medicine, Soonchunhyang University, Cheonan 31511, Republic of Korea; yjyjlee37@naver.com (Y.L.); m1037624@sch.ac.kr (S.-H.L.); dongsicchoi@gmail.com (D.C.)
2 Division of Molecular Cancer Research, Soonchunhyang Medical Research Institute, Soonchunhyang University, Cheonan 31511, Republic of Korea
3 Department of Tropical Medicine, College of Medicine, Soonchunhyang University, Cheonan 31511, Republic of Korea; namhs@sch.ac.kr (H.-S.N.)
4 Department of Dermatology, Soonchunhyang University Hospital, Seoul 04401, Republic of Korea; 132351@schmc.ac.kr (K.D.K); 146079@schmc.ac.kr (M.H.C)
* Correspondence: mkcho@schmc.ac.kr (M.-K.C.); leeyj@sch.ac.kr (Y.-J.L.); Tel.: +82-41-570-2443
† These authors contributed equally to this article.
* Correspondence:
Yoon-Jin Lee, PhD
Department of Biochemistry, College of Medicine,
Soonchunhyang University, Cheonan 31511, Republic of Korea
Phone: +82 41 570 2443
E-mail: leeyj@sch.ac.kr
|
Reviewer 1
Comment 1: 1. Abstract:
Response 1: Thank you for your valuable feedback. We incorporated your feedback by shortening and simplifying sentences in the abstract to improve readability. In addition, we clarified the study objectives, explicitly stating the reported metabolic differences between the two cell lines and the rationale for using these two distinct melanoma cell lines.
We've taken your feedback into consideration and made the following revisions:(line 18-32)
Comment 2: 2. Introduction: While the introduction provides valuable context, it is at times overly detailed. It could be shortened by removing some of the extensive epidemiological information and focusing more directly on the main objective of the study, which should be clearly articulated at the end of the section.
Response 2: Thank you for your valuable feedback. In the Introduction, we reduced the amount of epidemiological information and revised the text to present the main research objectives more directly, placing the study aims at the end of the section (line 37-44).
|
|
|
|
Comment 3: 3. Methods: -A clear statement regarding the number of replicates for each assay should be added directly in the experimental descriptions. Currently, this information appears only in the statistical section or in figure legends, which may make the methodology harder to follow. -Although the level of detail demonstrates precision, the methodological section is very lengthy. For consideration: certain parts of manufacturers’ instructions (e.g., kit procedures, the formula for calculating hexokinase activity) could be summarized or replaced with a reference such as “according to the manufacturer’s protocol,” as is common practice in publications.
Response 3: Thank you for your valuable feedback. -We clearly specified the number of replicates performed in the experiments to facilitate a better understanding of the methodology (lines 113-281).
-Detailed descriptions of the methods were provided, with certain parts of the manufacturer’s protocols summarized (lines 113-281).
Comment 4: 4. Resluts: It would be helpful to include comments on the statistical significance of the main differences observed (e.g., p-values or significance thresholds). The only minor terminological issue concerns the use of the term “necrosis” in the title of section 3.4, whereas the study actually investigated necroptosis. Using the term “necroptosis” would avoid confusion with accidental necrosis. This does not affect the conclusions but is worth correcting in the final version. For Figure 2C, please provide higher-quality microscopic images. Please clarify why, after determining the IC50 values in the cytotoxicity assessment, these values were not used for subsequent experiments and why different concentrations of RSV were selected instead.I also recommend moving the wound healing assay results to the Results section and removing them from the Discussion.
Response 4: Thank you for your valuable feedback. -In the Results section and figure legends, we have included references to statistical significance, including p-values or significance levels, for all key differences.
-To avoid confusion with accidental necrosis, we have consistently used the term "necroptosis" (line 393).
- The microscopy images in Figure 2C, including those of normal cells (HEMn-MP), have been enhanced for improved resolution (line 339, Figure 2).
-The rationale for selecting the different RSV concentrations has been added to lines 316–318, and the wound healing assay results have been moved to the Results section (line 448).
|
|
|
|
In addition, issues related to English language quality, as well as grammatical errors and typos mentioned by the reviewer, have also been corrected.
This manuscript has not been submitted in whole or in part to any other journal, and its English language editing was completed by an English proofreading company (HARRISCO).
|
|
|
|
|
Reviewer 2 Report
Comments and Suggestions for Authors
The manuscript entitled “Resveratrol Targets Glycolytic Enzymes HK II and PKM2 to 2 Promote Concurrent Apoptotic and Necrotic Cell Death in Malignant Melanoma” investigates the effects of resveratrol on hexokinase II and pyruvate kinase M2 in two malignant melanoma cell lines, G361 and SK-MEL-24. It reveals the selectivity of the treatment between two different cell lines, while also identifying possible novel markers for the selection of responsiveness to melanoma therapy. While providing interesting findings, several points need to be improved.
Specific comments:
Title:
Please, remove the full stop.
Introduction
- Lines 57 and 60: Provide full names for PD-1, CTLA-4 and BRAF.
- Is there a metabolic difference described between the two cell lines? Provide a rationale for the aim of the study for using two different melanoma cell lines.
- The experiments on the human tissue are not mentioned in the aim. Outline the aim of why these experiments were performed, to assess the translational and clinical relevance? Also, add that comparison with normal melanocytes was performed, and the metabolic differences between the two malignant cell lines to justify the aim.
Methods
- Line 110: Were these specimens the excised melanoma? Was there adjunctive tissue included? The use of human tissue specimens was not mentioned in the aim. How many samples were used? Please indicate in detail how many samples were used, the histological type of melanoma, and disease stage, as this is important for data interpretation.
- Explain in brief why the normal melanocyte cell line was not used in subsequent experiments, after determining cell viability. Perhaps a short paragraph that explains the study design could be enough for better understanding.
- Lines 159, 160 and 170: Please add the producer and full name of the microscope, used magnification, camera type and used software for the photomicrographs analysis.
- Lines 167-169: Authors state that the cells were grown in the medium supplemented with 5% For wound healing assay, the cells should be deprived of FBS to prevent its effects on cell proliferation and migration. Please, correct this.
- Line 170: What equation was used to calculate the percentage of the wound closure?
- Line 178: After slides fixation, were they dehydrated in a series of ethanol prior to staining? Add this information to this sentence.
- HK activity: Provide information on which samples were HK activity measured? Cell lines or human tissue? The method for PKM2 activity determination is missing.
Results
- Line 275: Explain in the methods on which samples the enzyme activity is measured.
- The order of methods doesn't follow the order of the presented results, so it is difficult to follow. To obtain logical order and improve clarity, try to present methods and results consistently, following the same order.
- Line 278: It is stated in the methods that ANOVA test was used, while in the results Mann-Whitney U test was mentioned. Please, correct and be consistent.
- Figure 2. Provide images with higher magnification, as these morphological changes are not quite visible when presented like this. Also, provide photomicrographs of the normal cells treated with the same RSV concentrations.
Discussion
- Lines 435-439: Provide references for these two sentences.
- Figure 6. Please add a scale bar to the photomicrographs, and move the figure to the Results section.
- Lines 459-462: Provide references for these sentences.
- How would the authors explain the different responses to RSV by two different melanoma cell lines? Is there available literature data that describes metabolic differences between two different cell lines?
- Lines 512-513: Provide reference.
The conclusion is overall well written and supports the results of the study. However, the sentence: ”Furthermore, G361 555 and SK-MEL-24 cell antagonists induced massive left-right motility in response to RSV.” is not clear and needs to be rephrased.
General recommendation:
This research provides valuable information on metabolic differences between different melanoma cells that could influence therapeutic outcome after RSV treatment in vitro, and could be significant for possible clinical translation and identification of novel prognostic and therapeutic markers. It could be interesting to a broad readership, but several issues need to be corrected before publication. I suggest that it could be accepted after a minor revision.
Author Response
Dear Reviewers:
We sincerely thank you for the opportunity to revise our manuscript and appreciate the time and effort the reviewers have dedicated to evaluating our work.
The reviewers’ comments were very helpful in improving the clarity and quality of our manuscript. We have carefully addressed all the suggestions and made revisions accordingly. We believe that these changes have substantially improved the manuscript, and we hope that the revised version is now suitable for publication in Current Issues in Molecular Biology.
All modifications have been made in accordance with the reviewers’ recommendations, and the revised sections have been highlighted in blue for clarity.
Thank you for your consideration.
Sincerely yours,
Yoon-Jin Lee, PhD
Department of Biochemistry, College of Medicine,
Soonchunhyang University, Cheonan 31511, Republic of Korea
Phone: +82 41 570 2443
E-mail: leeyj@sch.ac.kr
Point-by-Point Responses to Editor
Manuscript ID: cimb-4015648
Title: Resveratrol Targets Glycolytic Enzymes HK II and PKM2 to Promote Concurrent Apoptotic and Necrotic Cell Death in Ma-lignant Melanoma
Authors: Yeji Lee 1, Sang-Han Lee 1,2, Dongsic Choi 1,2, Hae-Seon Nam 2,3, Ki Dam Kim 4, Min Hyuk Choi 4, Moon-Kyun Cho 2,4*,† and Yoon-Jin Lee 1,2*,†
1 Department of Biochemistry, College of Medicine, Soonchunhyang University, Cheonan 31511, Republic of Korea; yjyjlee37@naver.com (Y.L.); m1037624@sch.ac.kr (S.-H.L.); dongsicchoi@gmail.com (D.C.)
2 Division of Molecular Cancer Research, Soonchunhyang Medical Research Institute, Soonchunhyang University, Cheonan 31511, Republic of Korea
3 Department of Tropical Medicine, College of Medicine, Soonchunhyang University, Cheonan 31511, Republic of Korea; namhs@sch.ac.kr (H.-S.N.)
4 Department of Dermatology, Soonchunhyang University Hospital, Seoul 04401, Republic of Korea; 132351@schmc.ac.kr (K.D.K); 146079@schmc.ac.kr (M.H.C)
* Correspondence: mkcho@schmc.ac.kr (M.-K.C.); leeyj@sch.ac.kr (Y.-J.L.); Tel.: +82-41-570-2443
† These authors contributed equally to this article.
* Correspondence:
Yoon-Jin Lee, PhD
Department of Biochemistry, College of Medicine,
Soonchunhyang University, Cheonan 31511, Republic of Korea
Phone: +82 41 570 2443
E-mail: leeyj@sch.ac.kr
|
Reviewer 2
Comment 1: Introduction: 2. Is there a metabolic difference described between the two cell lines? Provide a rationale for the aim of the study for using two different melanoma cell lines. 3. The experiments on the human tissue are not mentioned in the aim. Outline the aim of why these experiments were performed, to assess the translational and clinical relevance? Also, add that comparison with normal melanocytes was performed, and the metabolic differences between the two malignant cell lines to justify the aim.
Response 1: Thank you for your valuable feedback. 1. The full names of PD-1, CTLA-4 (lines 50-51), and BRAF (lines 53-54) have been provided.
2. We have described the reported metabolic differences between the two cell lines, including relevant references [22], and provided the rationale for using these two distinct melanoma cell lines in lines 94–104. [22] Fisher G.M.; Gopal, Y.N.V.; McQuade, J.L.; Peng, W.; DeBerardinis, R.J.; Davies, M.A. Metabolic strategies of melanoma cells: Mechanisms, interactions with the tumor microenvironment, and therapeutic implications. Pigment Cell Melanoma Res. 2018, 31(1), 11–30.
3. We have added information regarding human tissues, specifically noting the comparison with normal melanocytes, and highlighted that the metabolic differences between the two malignant cell lines support the study objectives (lines 105-112).
Comment 2: Methods: 1. Line 110: Were these specimens the excised melanoma? Was there adjunctive tissue included? The use of human tissue specimens was not mentioned in the aim. How many samples were used? Please indicate in detail how many samples were used, the histological type of melanoma, and disease stage, as this is important for data interpretation. 2. Explain in brief why the normal melanocyte cell line was not used in subsequent experiments, after determining cell viability. Perhaps a short paragraph that explains the study design could be enough for better understanding. 3. Lines 159, 160 and 170: Please add the producer and full name of the microscope, used magnification, camera type and used software for the photomicrographs analysis. 4. Lines 167-169: Authors state that the cells were grown in the medium supplemented with 5% For wound healing assay, the cells should be deprived of FBS to prevent its effects on cell proliferation and migration. Please, correct this. 5. Line 170: What equation was used to calculate the percentage of the wound closure? 6. Line 178: After slides fixation, were they dehydrated in a series of ethanol prior to staining? Add this information to this sentence. 7. HK activity: Provide information on which samples were HK activity measured? Cell lines or human tissue? The method for PKM2 activity determination is missing.
Response 2: Thank you for your valuable feedback. 1. The purpose of using human tissues is described in the study objectives at the end of the Introduction (lines 108–109), and detailed information about the tissues used has been provided in the Materials and Methods, Section 2.1 (lines 114-126).
2. After assessing the viability of normal melanocyte cell lines, their intended purpose was described in the Cell Culture section of the Materials and Methods (lines 146–148) and in the Results section (lines 316-318).
3. Information on the microscopes used, including manufacturer, model, camera type, and image analysis method, has been added to Section 2.6 of the Materials and Methods (lines 182–183), and scale bars and magnification details have been included in the figure legends.
4. A condition without FBS was included to prevent effects on cell proliferation and migration, and for clarity, the data were graphed based on migration distance to avoid reader confusion (Figure 6).
5. To avoid reader confusion, the wound closure rate was expressed using the raw distance values while setting the control as 100% (Figure 6).
6. It was specified in Section 2.6 of the Materials and Methods (lines 179-181) that the slides were dehydrated using an ethanol step prior to staining after fixation.
7. Measurement of HK activity in cell lines was described in Section 2.11 (line 243) of the Materials and Methods, while PKM2 activity was not assessed in this study.
|
|
|
|
Comment 3: Results: 1. Line 275: Explain in the methods on which samples the enzyme activity is measured. 2. The order of methods doesn't follow the order of the presented results, so it is difficult to follow. To obtain logical order and improve clarity, try to present methods and results consistently, following the same order. 3. Line 278: It is stated in the methods that ANOVA test was used, while in the results Mann-Whitney U test was mentioned. Please, correct and be consistent. 4. Figure 2. Provide images with higher magnification, as these morphological changes are not quite visible when presented like this. Also, provide photomicrographs of the normal cells treated with the same RSV concentrations.
Response 3: Thank you for your valuable feedback. 1. The measurement of enzyme activity in cells was described in Sections 2.10–2.12 of the Materials and Methods.
2. The Materials and Methods sections were reorganized to match the sequence of the Results, ensuring that the description of each experiment corresponds to its presentation in the Results. 3. The analyses were revised using the ANOVA method, and content that could cause confusion was removed.
4. Images of normal cells (HEMn-MP) treated with the same concentrations of RSV were provided, and all cell images were inserted at 300 dpi resolution with 400× magnification (Figure 2C, line 337).
Comment 4: Discussion: 1. Lines 435-439: Provide references for these two sentences. 2. Figure 6. Please add a scale bar to the photomicrographs, and move the figure to the Results section. 3. Lines 459-462: Provide references for these sentences. 4. How would the authors explain the different responses to RSV by two different melanoma cell lines? Is there available literature data that describes metabolic differences between two different cell lines? 5. Lines 512-513: Provide reference.
Response 4: Thank you for your valuable feedback. 1. A corresponding reference for this statement has been added as [23]. [23] Shen D.; Zhang, L.; Li, S.; Tang, L. Metabolic reprogramming in melanoma therapy. Cell Death Discov. 2025, 11(1), 308.
2. A scale bar has been added, and the figure has been relocated to the Results section (lines 451-453).
3. A corresponding reference for this statement has been added as [24]. [24] Wang, G.; Lai, Y.; Chen, X.; Li, N., Zhong, C.; Yan, Y.; Ma, Q.; Hong , X.; Zhu, N.; Yu, W. Hexokinase 2 promotes tumor development and progression. Am J Cancer Res. 2025, 15(10), 4499–4515.
4. An explanation for the differential responses of the two melanoma cell lines to RSV, supported by previously published literature on their metabolic differences, has been added and is presented in lines 500–504.
5. A corresponding reference for this statement has been added as [38]. [38] Li, Y.; Wu, C.; Shah, S.S.; Chen, S.; Wangpaichitr, M.; Kuo, M.T.; Feun, L.G.; Han, X.; Suarez, M.; Prince, J.; Savaraj, N. Degradation of AMPK-α1 sensitizes BRAF inhibitor-resistant melanoma cells to arginine deprivation. Mol Oncol. 2017, 11(12), 1806–1825.
Comment 5: Conclusion: The conclusion is overall well written and supports the results of the study. However, the sentence: ”Furthermore, G361 555 and SK-MEL-24 cell antagonists induced massive left-right motility in response to RSV.” is not clear and needs to be rephrased.
Response 4: Thank you for your valuable feedback. The term ‘left-right motility’ has been replaced with ‘migration capacity’ in line 548. |
|
|
|
In addition, issues related to English language quality, as well as grammatical errors and typos mentioned by the reviewer, have also been corrected.
This manuscript has not been submitted in whole or in part to any other journal, and its English language editing was completed by an English proofreading company (HARRISCO).
|
|
|
|
|